# Prediction of Protein–Protein Interaction Sites Using Convolutional Neural Network and Improved Data Sets

**DOI:** 10.3390/ijms21020467

**Published:** 2020-01-11

**Authors:** Zengyan Xie, Xiaoya Deng, Kunxian Shu

**Affiliations:** Chongqing Key Laboratory of Big Data for Bio Intelligence, Chongqing University of Posts and Telecommunications, Chongqing 400065, China; deng_xiaoya@outlook.com

**Keywords:** protein–protein interaction sites, convolutional neural network, residue binding propensity

## Abstract

Protein–protein interaction (PPI) sites play a key role in the formation of protein complexes, which is the basis of a variety of biological processes. Experimental methods to solve PPI sites are expensive and time-consuming, which has led to the development of different kinds of prediction algorithms. We propose a convolutional neural network for PPI site prediction and use residue binding propensity to improve the positive samples. Our method obtains a remarkable result of the area under the curve (AUC) = 0.912 on the improved data set. In addition, it yields much better results on samples with high binding propensity than on randomly selected samples. This suggests that there are considerable false-positive PPI sites in the positive samples defined by the distance between residue atoms.

## 1. Introduction

Proteins play key roles in various aspects of life [1] by physically interacting with other proteins [2,3]. Protein–protein interactions (PPIs) are the molecular basis for many biological processes, such as signal transduction, transport, metabolism, gene expression, growth and proliferation of cells [4,5]. Protein-binding interfaces are heterogeneous and some interface residues contribute more to binding than the others. These residues are called “hotspots” [6,7,8,9,10]. Hotspots are often pre-organized in the unbound protein state. So it is suggested that much of the protein surface does not accommodate binding and the potential binding sites of a protein are already imprinted in its unbound state [8].

PPI sites are critical domains for selective recognition of molecules and the formation of complexes [11]. Identification of protein interaction sites is pivotal for understanding protein function, elucidating signal transduction networks, and drug design [12]. Experimental methods such as NMR and X-ray crystallography have been used to characterize PPI sites [13,14]. However, these techniques are expensive and time-consuming [15,16]. With the fast expansion of resolved sequences and structural data of proteins, several kinds of computational methods such as molecular dynamics methods [17,18] and machine learning methods have been proposed to predict PPI sites. Among these methods, machine-learning methods are most successful, using the following models [14,19]: support vector machine (SVM) [20,21,22,23,24,25] and fuzzy SVM [26], neural networks (NN) [27,28,29,30,31,32], Bayesian networks (BN) [33,34], naive Bayes classifier (NBC) [35,36], random forests (RF) [12,37,38,39], cascade random forests (CRF) [40], conditional random fields (CRF) [41], extreme learning machine (ELM) [42], L1-logreg classifier [43], and the ensemble method [14,15,44,45]. As a recent development of neural networks, deep-learning is a rapidly growing branch of machine learning and has also been used to predict PPI sites [19]. As a representative algorithm of deep learning, the convolutional neural network (CNN) [46,47] can be used for representation learning and extracting high-order features from the input information.

Popular PPI sites prediction methods can be sorted into three groups according to the information they are based on.

Sequence-based methods. Methods based on sequence information use features extracted from protein sequences to predict protein interaction sites. PPiPP [48] uses the position-specific scoring matrix (PSSM) and amino acid composition to predict PPI sites and achieves an area under the receiver operating characteristic (ROC) curve (AUC) of 0.729. DLPred [19], which uses long-short term memory (LSTM) to learn features such as PSSM, physical properties, and hydropathy index, obtains a higher AUC score of 0.811. Still, we need more information to improve prediction accuracy.Structure-based methods. Knowledge of the three-dimensional (3D) structure of the protein complex provides much valuable information on the protein interaction sites [14]. Some PPI sites predictors utilize 3D structural information of proteins for prediction. ProMate combines all the significant interface properties and reaches a success rate of 0.70 [33]. Bradford and Westhead [21] achieved a successful prediction rate of 0.76 based on protein structure data.Methods based on integrated information. Three-dimensional structure of proteins are far more difficult and expensive to elucidate than protein sequences, so its magnitude in protein structure databases such as the Protein Data Bank (PDB) [49] is remarkably smaller compared to that of sequences in protein sequence databases like UniProt [50]. Therefore, most methods use a combination of structural and sequence information for the prediction of PPI sites. Li et al. [38] use physicochemical properties, sequence conservation, residue disorder, secondary structure, solvent accessibility, and five 3D structural features to train a random forest model to predict PPI sites. SPPIDER [51] uses relative solvent accessibility (RSA), sequence and structure features to predict PPI sites and demonstrates that RSA prediction-based fingerprints of protein interactions significantly improve the discrimination between interacting and noninteracting sites. It yields an overall classification accuracy of about 0.74 and Matthews correlation coefficients (MCC) of 0.42. IntPred [39] uses 11 features of both sequence and structure and obtains a specificity of 0.916 and a sensitivity of 0.411. PAIRpred [24] captures sequence and structure information about residue pairs through pairwise kernels that are used for training a support vector machine classifier. This method gives a remarkable AUC score of 0.870 and rank the first positive prediction (RFPP) value on the 176 complexes in protein–protein docking benchmark version 4.0 (DBD 4.0) [52] with its structure kernel.

In this study, we propose a novel statistics-based method for judging the binding propensity of amino acids and apply it to the partitioning of samples. We extracted the sequence and structure features of each sample and input them into the convolutional neural network for training. Compared to previous methods, our approach has made significant improvement in the AUC score (0.912) and some of the RFPP values (RFPP (100) = 580) on the 116 dimers in DBD 4.0 [52].

## 2. Results

### 2.1. Distribution Tendency of Residues in Proteins

In order to demonstrate the distribution tendency of residues, we first compared the abundance of residues (AR) between the protein surface (AR_s_) and whole protein (AR_w_) and used AR_w_/AR_s_ as the indicator of the tendency of a residue to be inside of proteins (Table 1, Section 4.3).

From Table 1, we find that most hydrophobic residues (alanine, leucine, isoleucine, valine, glycine, cysteine, phenylalanine, proline) and amphipathic residues (tryptophan, tyrosine, and methionine) tend to distribute inside proteins except proline which tends to be on the protein surface, and glycine which shows a weak surface tendency. Charged (arginine, lysine, aspartic acid, and glutamic acid) and hydrophilic residues tend to appear on the protein surface with the exception of histidine, which shows no tendency towards protein inside or surface.

### 2.2. Residue Binding Propensity

Protein residues exhibit different binding propensity for different residues in protein–protein interfaces. We used a statistical method (Section 4.4) to classify residues interacting with one certain residue into high and low binding propensity residue groups and compared the binding propensity with polarity, hydrophobicity, and distribution tendency of residues. 

Table 2 shows the relative abundance of interacting residues (RAIR, Section 4.4) which indicates the binding propensity of each residue with all 20 residues. Data of abundance of interacting residues (AIR) and AR are available in Appendix A, respectively. Polarity, hydrophobicity [45], and the ratio of AR_w_ and AR_s_ from Table 2 are also placed in the same table for comparison. 

From Table 2, we find that (1) Ten residues (leucine, isoleucine, valine, arginine, histidine, cysteine, methionine, tyrosine, tryptophan, and phenylalanine) show a high propensity to bind to most residues (RAIR scores ≥ 1, shaded), while the other ten residues show a low binding propensity. (2) Most of the residues with high binding propensity overlap with the residues with polarity ≤ 7 (shaded polarity scores) except arginine (polarity = 10.5) and histidine (polarity = 10.4). (3) Residues with positive hydrophobicity (shaded hydrophobicity scores) also exhibit higher binding propensity but with more exceptions. Alanine, glycine, and proline have positive hydrophobicity but low binding propensity. On the contrary, arginine and histidine have negative hydrophobicity but high binding propensity. (4) Interestingly, residues with AR_w_/AR_s_ ≥ 1 (shaded AR_w_/AR_s_ scores) show high coincidence with those with high binding propensity, only with the exception of alanine (AR_w_/AR_s_ = 1.22) and arginine (AR_w_/AR_s_ = 0.95).

Finally, a total of 221 pairs of residues with high binding propensity (shaded RAIR scores in row 2–21 in Table 2) were obtained and used for further screening of positive samples. Each residue-pair contact propensities in a protein–protein interface are shown in Appendix A.

### 2.3. Positive Samples with High Binding Propensity

In order to verify the effectiveness of the improved positive samples, we used the same model parameters (Section 4.6) to perform a leave-one-complex-out cross-validation verification for two sample data sets, one with high binding propensity and another with no propensity.

According to the definition of interacting residue pairs (Section 4.2) from PAIRPred [24] and PPiPP [48], a total of 12,138 positive and 5,534,983 negative samples from 138 dimers of the DBD 5.0 version [53] were obtained (Section 4.1). Among the positive samples, 6739 residue pairs with binding propensity ≥ 1 were used as final positive samples in this study. There is an average of 49 pairs of positive samples for each dimer.

Test data sets used in this study were extremely imbalanced, so we used AUC as the main measure to evaluate the model performance on two data sets. The AUC for data set with no propensity is 0.824, while the AUC for data set with high propensity reaches 0.912 (Figure 1). We also provided accuracy and recall under different thresholds in Appendix A.

### 2.4. Comparison with Randomly Sampled Data Set

To further verify the rationality of the binding propensity, we conducted a five-fold cross-validation to compare the performance of our model on data sets with high binding propensity and data sets randomly sampled (also have 6739 residue pairs) from original positive samples by using 138 dimers from DBD 5.0 [53]. 

There is a significant difference between the AUCs of these two data sets. The AUC for the data set with high propensity is 0.889 ± 0.007 (Figure 2a), while the AUC for the randomly sampled data set is 0.811 ± 0.006 (Figure 2b). It is notable that the result of five-fold cross-validation is close to that of leave-one-complex-out cross-validation (0.912) on the data set with high propensity. This result indicates that our model benefits from the data set screened using a propensity score and identifies PPI sites with more accuracy.

### 2.5. Comparison with Existing Methods

To further evaluate our model, we compared its performance with those of PSIVER [35], PPiPP [48], SSWRF [54], DLPred [19], and PAIRPred [24] using AUC scores (Table 3). The first four methods used sequence-based features, while PAIRPred and our method used both sequence- and structure-based features. The AUC score of our method is noteworthily higher than that of the other methods. The results also prove the importance of structural features in PPIs site prediction.

PAIRPred [24] is currently one of the best-performing methods for predicting protein interaction sites based on sequence and structure features. It uses SVM with pairs of kernels to predict if two residues interact with each other. We compared the performance of PAIRPred and our model on 116 dimers from DBD 4.0. The results are shown in Figure 3. We performed leave-one-complex-out cross-validation on positive samples with high propensity and obtained an AUC of 0.912, which is higher than that of PAIRPred (0.862). 

We also evaluate our model by using the first rank of the first positive prediction (RFPP, Section 4.7). For RFPP, our method performs better on 90% (169) and 100% (580) than PAIRPred (194 and 2861, respectively) for dimers in DBD 4.0, while PAIRPred has better RFPP results on 10%, 25%, 50%, and 75% (Table 4). Our method has significantly improved on RFPP (100) of the complexes, which means that our model has better generalization ability.

## 3. Discussion

Protein–protein interactions play essential roles in many biological processes. Among different methods proposed to predict PPIs, machine learning is the most promising and commonly used algorithm. Deep learning is a popular machine learning branch and has been applied to many fields in recent years. In this paper, we used a convolutional deep learning model and improved data sets for the prediction of PPI sites and obtained a result of AUC 0.912, which is better than those of published predictors.

The protein–protein docking benchmark data sets (DBD) have been widely used for evaluation of PPIs prediction methods. Its latest version is 5.0 [53]. We emphasized comparison with PAIRpred [24] since it is one of the best-performing PPI predictors. PAIRpred was tested on DBD 4.0 [52], but SPINE X [55], on which PAIRpred was dependent on, is obsolete now, so we cannot calculate the AUC and RFPP of PAIRpred on DBD 5.0. As a result, we compared the results of PAIRpred and our method on 116 dimers in DBD 4.0, and used 138 dimers in DBD 5.0 (including 116 dimers in DBD 4.0) to further evaluate our model.

Residue interface propensities were observed in different kinds of protein complexes [25] and has been used to improve prediction accuracy of PPI sites in different studies [4,21,25,33,56,57]. It is usually used as a parameter of predicting models. In this study, we used residue binding propensity to screen positive samples and improved the performance of prediction remarkably. Our method may be a little radical, but the result suggests that it makes sense to reduce the fraction of false positive samples by introducing binding propensity.

It is found that polar residues are statistically disfavoured in interface sites, with the exception of arginine [25,33,56]. In order to show the correlation between polarity, hydrophobicity, and binding propensity in a more intuitive way, we compared polarity, hydrophobicity [45], and RAIR, and found that besides arginine, another polar residue histidine also exhibited a high binding propensity. This coincides with the result of another research [58] which found that histidine was favoured in all types of interactions.

An interesting finding of this study is that residues which tend to be inside of proteins (AR_w_/AR_s_ > 1) have higher binding propensity. This seems strange but makes sense since most of these residues are hydrophobic and if they appear at the surface of proteins they tend to interact with the hydrophobic residues on the surface of other proteins. An exception is alanine whose side chain is just a methyl, which disfavours interaction with other residues and has been utilized for alanine-scanning mutagenesis analysis [9]. On the contrary, residues with large hydrophobic side chains, such as tryptophan, was found to have a unique role in the folded structure and the binding sites of proteins [59]. Charged residues show high binding propensity for oppositely charged residues. Arginine was found to be the most frequently occurring residue in known protein interaction sites because of its wide radius of action [11]. We also found that arginine exhibited a high binding propensity in our study, although it tended to appear at the protein surface (AR_w_/AR_s_ = 0.74).

In conclusion, our convolutional deep learning model performs well for prediction of protein interaction sites, especially on the improved data set with high binding propensity. This suggests that a nonnegligible portion of false positive interacting pairs exist in the original positive samples obtained by 6Å definition, which may impede the efforts of improving the accuracy of prediction for PPI sites. Reducing false positive interacting samples is likely to become a promising direction for PPI site prediction studies.

## 4. Materials and Methods 

### 4.1. Data Sets

The protein–protein docking benchmark data set (DBD, version 5.0) [53] and DBD 4.0 [52] were used in this work. DBD 5.0 contains 139 non-redundant dimers with characterized bound and unbound X-ray crystallography structures. DBD 4.0 [52] contains 174 complexes among which 116 are dimers and form a subset of DBD 5.0. Two interaction protein chains of a dimer are from different families defined by Structural Classification of Proteins (SCOP) with sequence identity less than 30% [53]. There are a few deletions in the sequence of 1ZLI in unbound state so it is excluded from the data sets. Finally, 174 complexes from DBD 4.0 were used for computing of residue distribution tendency and statistics of binding propensity, 116 dimers from DBD 4.0 were used for model comparison, and 138 dimers from DBD 5.0 were used to further validate our model (Table 5 and Table 6).

### 4.2. Definition of Interacting Residue Pairs

A pair of residues from two proteins are considered to have interaction if the Euclidean distance between any two atoms from each of the two residues in the bound state is less than or equal to 6 Å [24,48]. According to this definition, 12,138 positive samples (interacting residue pairs) and 5,522,852 negative samples (non-interacting residue pairs) were obtained, each dimer has an average of 88 positive samples and 40,006 negative samples. Contacting residues within a protein chain were not included. 

The number of negative samples in this study is much larger than that of positive samples. This imbalanced data made it difficult to train the model and the under-sampling might lead to information loss, so we used the EasyEnsemble algorithm [60] to build the training set with equal positive and negative samples.

### 4.3. Distribution Tendency of Residues in Proteins

Residues show different preferences of locations in proteins. We calculated the abundance of residues (AR) of the protein surface (AR_s_) and whole proteins (AR_w_) using 174 complexes from DBD 4.0 (Section 4.1) and used AR_w_/AR_s_ as the indicator of a residue’s tendency to be inside or at the surface of proteins. If AR_w_/AR_s_ for a residue is larger than 1, it tends to be inside of proteins. Otherwise, if AR_w_/AR_s_ < 1, it appears more often at the suface of proteins.

DSSP [61] from xssp [62] was used to calculate the solvent-accessible surface area (ASA) of a residue. If the ratio of ASA of a residue to its maximum ASA (Table 7) [63,64] is larger than or equal to 0.16 [28,63], then it is defined as a surface residue.

### 4.4. Binding Propensity of Residue Pairs

The above definition of interacting residues is concise and easy to use. But binding propensity between residues varies considerably. Residues with strong binding propensity may dominate the interaction and lead their adjacent residues with weak binding propensity into the defined range of interacting residues. In this situation, the dominant interacting residues (DIRs) are true positive samples, while it is more reasonable to classify the passive interacting residues (PIRs) as false positive samples (Figure 4). 

The binding propensity between different residues can be computed based on interacting residue frequencies [56]. In this study, a statistical method was used to classify residues interacting with one certain residue into high and low binding propensity residue groups. Residues of 174 protein complexes from DBD 4.0 were used to estimate the residue binding propensity. 

Relative abundance of interacting residues (RAIR) was used to indicate the binding propensity of each residue pair. Abundance of residues (AR) represents the frequency of each residue (20 in total) in the total number of suface residues of 174 protein complexes from DBD 4.0. The abundance of interacting residues (AIR) represents the frequency at which each residue interacts with 20 residues (400 pairs in total). The RAIR between residue *i* and residue *j* is defined as follows:(1)$$ARi=\frac{Ni}{N}$$
(2)$$AIRij=\frac{Mij}{Mi}$$
(3)$$RAIRij=\frac{AIRij}{ARj}$$
where *N* is the total number of all surface residues of 174 protein complexes, *N_i_* is the number of residue *i*, *M_ij_* is the number of residue *j* interacting with residue *i*, and *M_i_* is the total number of all residues interacting with residue *i*. 

RAIR is used for further classification of samples in this study. A pair of residues is considered to have a high binding propensity when RAIR ≥ 1 otherwise the pair has a low binding propensity.

### 4.5. Features

#### 4.5.1. Amino Acid Encoding

Twenty amino acids were coded as one-hot encoding [65] (Appendix A).

#### 4.5.2. Sequence Features

##### Profile Features

Position specific scoring matrix (PSSM) and position specific frequency matrix (PSFM) reflect the conservation of residues at specific positions of protein chains based on evolutionary information [3,24]. Each row of the PSSM or PSFM is a 20-dimensional vector. PSSM and PSFM were computed by running 3 iterations of PSIBLAST [66] against the NCBI NR database for a given protein with E-value set to 0.001. PSSM and PSFM columns were taken within a length 3 window centered at a residue of the protein to obtain a 3 × 40 matrix.

##### Amino Acid Physicochemical Properties

Twenty-four physicochemical properties of amino acids [67] are used in this study. Twenty amino acids are divided into three groups according to these properties and each group is encoded using one-hot encoding, thus each amino acid is represented as a 72-dimensional vector. e.g., alanine (A) is encoded as
(4)$$\begin{array}{l} A=[0,1,0,1,0,0,0,1,0,1,0,0,1,0,0,1,0,0,\\ 1,0,0,1,0,0,0,1,0,0,1,0,0,1,0,0,1,0,\\ 0,0,1,0,1,0,0,1,0,0,0,1,0,0,1,0,0,1,\\ 1,0,0,0,1,0,0,0,1,0,0,1,1,0,0,0,1,0] \end{array}$$

#### 4.5.3. Structure Features

Residues that play important roles in protein function generally appear at the surface of proteins. The accessible surface area (ASA) and the relative accessible surface area (RASA) were used to identify whether a residue is at the surface of a protein. The geometric properties of the protein surface can affect the interaction between proteins [38,68]. Protrusion index (CX) and depth index (DPX) were used to describe these properties. We also used hydrophobicity, which plays an important role in PPIs, protein folding and unfolding [69,70]. These five structure-based features were computed using PSAIA [71], which was developed for calculation of the geometric parameters of large protein structures and the prediction of protein interaction sites.

### 4.6. Deep Learning Model

The structure of the deep learning model used in this paper is shown in Figure 5. The source code of this study is available at https://github.com/Xiaoya-Deng/PPI-sites-prediction. 

#### 4.6.1. Input

Whether two residues interact is a binary classification problem. In our model, each sample is represented by ((*r_i_, l_i_*), *y_i_*), where (*r_i_, l_i_*) represents a pair of residues and *y_i_* is corresponding label. *y_i_*= 1 if two residues interact, otherwise *y_i_*= 0. 

Each residue pair is encoded as a 2 × 217 × 1-dimensional vector for the input of the network. 

#### 4.6.2. Convolutional Layers

Three convolutional layers are used in this paper. The filters of the first and the second convolutional layers have the size of 3 × 3 with depth of 32 and 64, respectively. The third convolutional layer filter has a size of 1 × 3 and a depth of 128. All three layers use stride 2 and no zero padding.

#### 4.6.3. Pooling Layers

There is a pooling layer after each convolutional layer. All the pooling layers use max pooling with filter size of 2 × 2. The strides for both directions are set to 2.

#### 4.6.4. Fully Connected Layer

The output of the last pooling layer is expanded into a 1-dimensional vector and used as the input of a fully connected neural network with 1024 neurons. Finally, a Softmax function is used as the classifier.

#### 4.6.5. Activation Function and Loss Function

Softplus and Softmax are used as the activation and loss functions, respectively, in our model.
(5)$$f(x)=\ln(1+e^{x})$$
(6)$$L(x_{i})=-\log\frac{e^{x_{i}}}{\sum {}_{j}e^{x_{j}}}$$

#### 4.6.6. Model Optimization

The AdamOptimizer in TensorFlow was used for training optimization. The dropout method and decay learning rate method were used to prevent over-fitting during training.

### 4.7. Performance Measure

A leave-one-complex-out cross-validation method [24] was used to evaluate our model. All positive and negative samples of one protein complex were chosen as the imbalanced validation set while the samples of the other complexes were used as the training set which consists of balanced positive and negative samples.

A probability value between 0 and 1 is returned after the samples are trained, while the class label is binary (1 for interaction and 0 for non-interaction). At a given probability threshold, any correctly predicted pair of interacting residues is designated as true positive (NTP), and any correctly predicted pair of non-interacting residues is designated as true negative (NTN). While false positive (NFP) and false negative (NFN) are pairs of residues that are incorrectly predicted to be positive or negative, respectively.

Accuracy, recall, and precision can be used for evaluation of machine learning models. But the balance between these parameters varies with thresholds. The other two evaluation methods are more commonly used: (1) the area under the ROC curve (AUC), where ROC is for a (1-specific) recall map that takes into account the entire threshold range; and (2) a set of optimal performance thresholds accuracy, recall and F1. In this study, we use AUC as the main performance metric.

For imbalanced data, AUC may give a false impression of accuracy [24]. So we employed another measure of accuracy proposed in PAIRpred [24], the first rank of the first positive prediction (RFPP). RFPP is defined as follows:(7)RFPP(p)=q

RFPP indicates *p*% of the dimers tested have at least one true positive interacting residue pair among the top *q* predictions [24].

### 4.8. Validation on Randomly Sampled Data

In order to further validate the rationality of a propensity score, we compared the result with propensity to that using randomly sampled interacting pairs from original positive samples. The same number of positive and negative samples (6739) were randomly selected from 12,138 original positive and 5,522,852 negative samples, respectively, using EasyEnsemble [60]. We used five-fold cross-validation to compare the AUCs of this random sample to those of the data set with high propensity to see if there is difference between them.

## Figures and Tables

**Figure 1 ijms-21-00467-f001:**
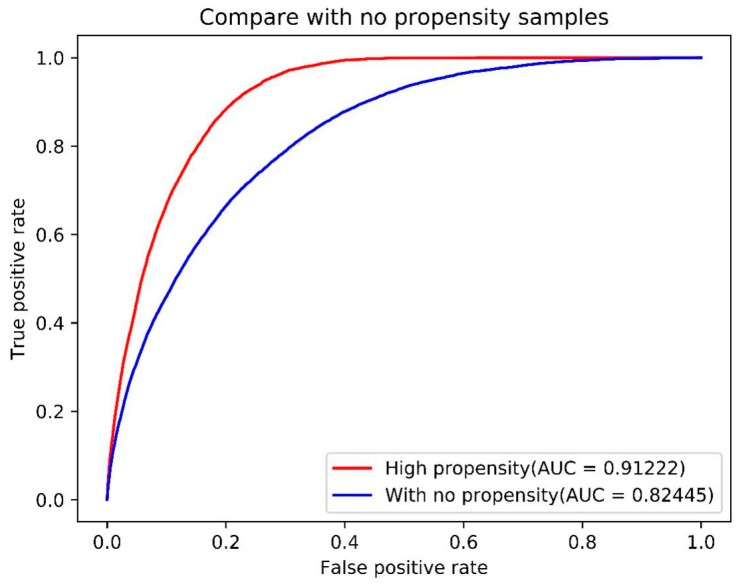
Receiver operating characteristic (ROC) curve and area under the curve (AUC) score of samples with high binding propensity and samples with no binding propensity.

**Figure 2 ijms-21-00467-f002:**
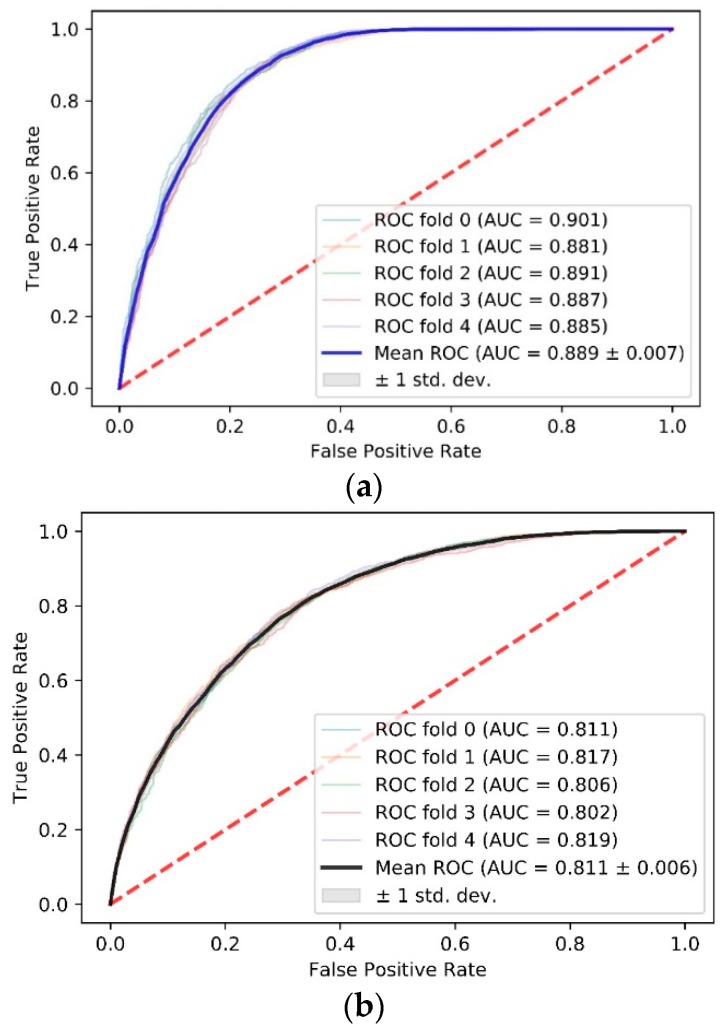
Comparison of AUC scores between data sets with high binding propensity and data sets randomly sampled. (**a**) ROC curve and AUC score of the high propensity data set using five-fold cross-validation; (**b**) ROC curve and AUC score of the random sampling data set using five-fold cross-validation. The red dotted line is a control line on which AUC = 0.5.

**Figure 3 ijms-21-00467-f003:**
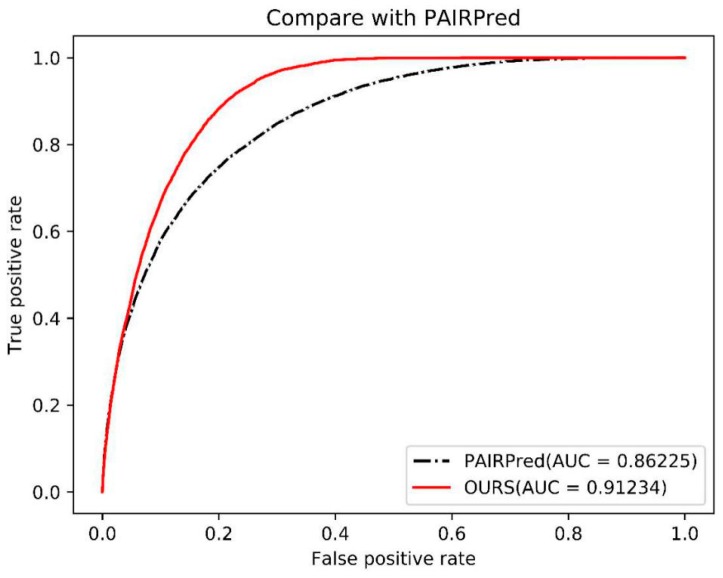
Compare with PAIRPred using AUC.

**Figure 4 ijms-21-00467-f004:**
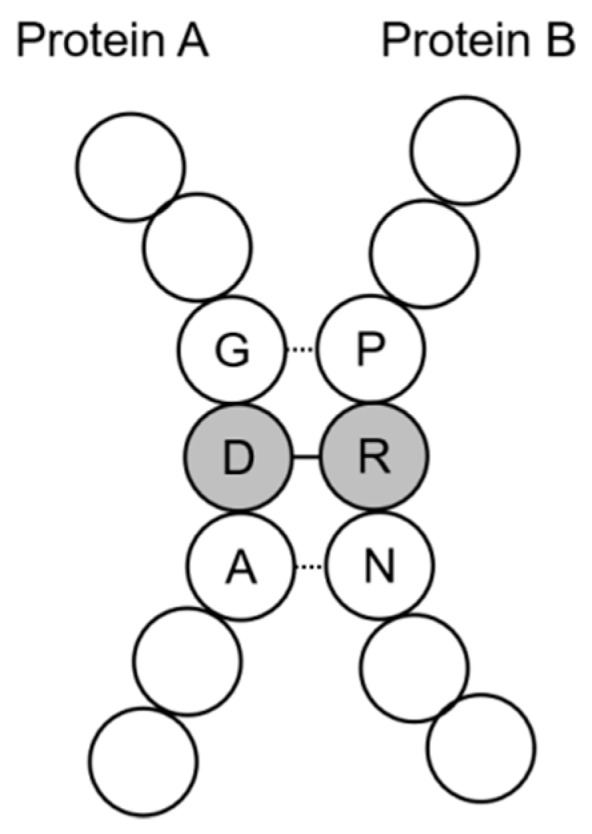
Diagram of interacting residue pairs between two proteins. Two false positive samples (G-P and A-N) caused by strong interaction (D-R). Due to the strong interaction of D and R, the Euclidean distance between G and P or A and N is less than 6.0 Å, but the interaction is dominated by D and R.

**Figure 5 ijms-21-00467-f005:**
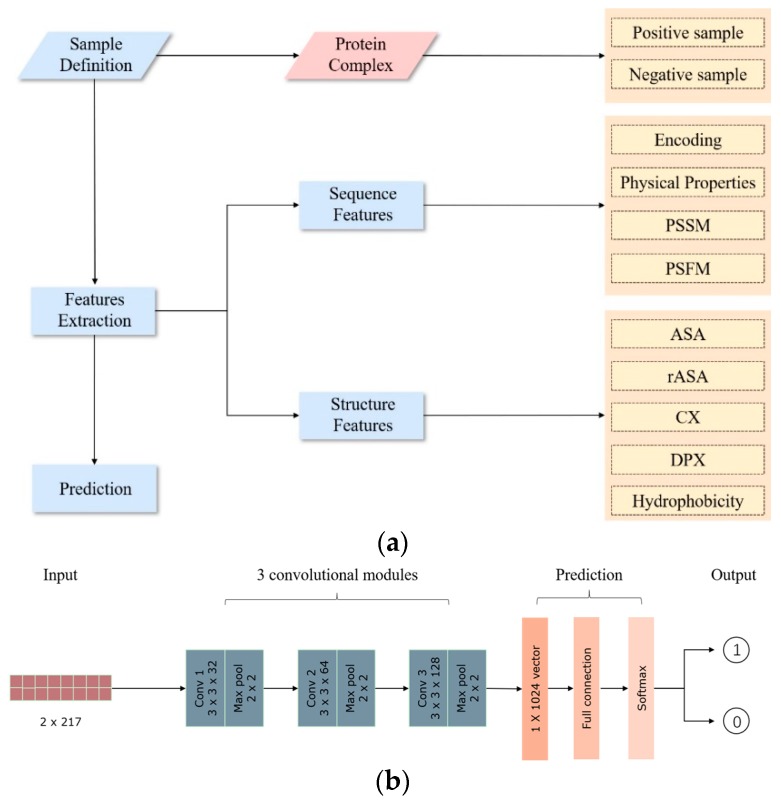
The process of prediction of PPI sites. (**a**) The overall flow chart for predicting pairs of interacting residues in this paper; (**b**) the structure of the convolutional neural network.

**Table 1 ijms-21-00467-t001:** Number and abundance of whole protein and surface residues.

Residue	N_w_	AR_w_	N_s_	AR_s_	AR_w_/AR_s_
A	9558	0.071	3599	0.058	1.22
L	11,587	0.087	2913	0.047	1.85
I	6700	0.050	1462	0.024	2.08
V	9407	0.070	2207	0.036	1.94
G	9930	0.074	4839	0.078	0.95
K	8155	0.061	5656	0.092	0.66
R	6093	0.046	3820	0.062	0.74
D	7561	0.057	4823	0.078	0.73
E	8667	0.065	5700	0.092	0.71
H	3128	0.023	1448	0.023	1.00
N	5812	0.043	3673	0.060	0.72
Q	5478	0.041	3396	0.055	0.75
S	9575	0.072	5498	0.089	0.81
T	8345	0.062	4069	0.066	0.94
C	2742	0.020	518	0.008	2.50
M	2785	0.021	791	0.013	1.62
Y	4811	0.036	1812	0.029	1.24
W	2035	0.015	552	0.009	1.67
F	5132	0.038	1101	0.018	2.11
P	6294	0.047	3845	0.062	0.76
Total	133,795	1	61,722	1	

N_w_: the number of specified amino acids in whole proteins; N_s_: the number of specified amino acids in the protein surface; AR_w_: abundance of residues of whole proteins; AR_s_: abundance of residues at the protein surface. AR_w_/AR_s_ ≥ 1 (shaded) represent residues that tend to distribute inside protiens.

**Table 2 ijms-21-00467-t002:** Relative abundance of interacting residues (RAIR) scores of residue pairs.

i\j	A	L	I	V	G	K	R	D	E	H	N	Q	S	T	C	M	Y	W	F	P
A	0.47	1.58	1.80	1.24	1.07	0.58	1.00	0.60	0.61	1.22	0.99	0.99	0.79	1.09	2.31	1.88	2.83	2.52	2.74	0.44
L	0.82	1.17	2.68	1.43	0.80	0.62	0.96	0.46	0.50	1.35	0.83	0.99	0.73	0.94	2.96	2.75	2.57	3.44	3.03	0.61
I	0.75	2.16	1.38	1.25	0.93	0.61	0.97	0.61	0.52	1.53	0.90	1.09	0.66	0.67	2.76	1.31	2.50	3.39	3.36	0.59
V	0.74	1.65	1.79	0.89	0.99	0.53	1.24	0.71	0.54	1.19	0.98	0.91	0.69	0.93	1.98	2.00	3.16	2.87	2.29	0.59
G	0.92	1.33	1.90	1.42	0.56	0.62	1.15	0.66	0.53	1.01	1.10	1.02	0.59	1.13	3.48	2.12	2.70	3.76	2.33	0.53
K	0.62	1.28	1.56	0.95	0.78	0.19	0.76	1.35	1.43	1.04	0.91	0.76	0.91	0.85	2.08	1.28	2.88	2.66	1.99	0.57
R	0.63	1.16	1.45	1.30	0.84	0.44	0.41	1.23	1.04	1.17	1.06	1.02	0.76	0.91	2.07	1.64	2.68	3.83	2.12	0.65
D	0.58	0.88	1.43	1.16	0.76	1.23	1.93	0.28	0.57	1.18	1.09	1.10	0.83	0.95	1.48	1.02	2.85	2.28	2.00	0.45
E	0.63	0.99	1.28	0.93	0.64	1.38	1.71	0.60	0.28	1.40	1.06	0.93	0.97	1.08	1.14	2.20	2.08	2.14	2.09	0.62
H	0.74	1.59	2.22	1.21	0.72	0.59	1.14	0.73	0.82	0.50	1.10	0.83	0.71	0.98	3.07	2.10	2.19	2.75	2.23	0.64
N	0.73	1.18	1.59	1.21	0.95	0.63	1.25	0.83	0.76	1.34	0.54	1.12	0.75	1.22	1.41	1.23	2.85	2.26	1.76	0.66
Q	0.75	1.46	1.99	1.16	0.90	0.54	1.25	0.86	0.69	1.04	1.16	0.42	0.75	1.07	1.35	1.62	2.37	2.46	2.43	0.80
S	0.77	1.37	1.54	1.12	0.67	0.82	1.19	0.82	0.91	1.14	0.99	0.95	0.37	1.05	3.53	2.00	2.12	2.92	2.16	0.68
T	0.82	1.36	1.19	1.16	0.98	0.59	1.08	0.73	0.78	1.20	1.23	1.04	0.81	0.51	1.10	1.68	2.85	3.61	2.21	0.66
C	0.76	1.88	2.17	1.09	1.34	0.64	1.09	0.50	0.37	1.67	0.63	0.58	1.20	0.48	0.63	2.49	2.36	2.78	3.09	0.63
M	0.74	2.08	1.23	1.32	0.97	0.47	1.03	0.41	0.84	1.36	0.65	0.83	0.81	0.88	2.97	1.19	2.54	1.86	3.80	0.73
Y	0.80	1.41	1.70	1.50	0.89	0.76	1.22	0.83	0.57	1.03	1.10	0.88	0.62	1.08	2.03	1.84	0.70	3.03	2.82	0.91
W	0.62	1.63	1.99	1.18	1.08	0.61	1.51	0.57	0.51	1.11	0.75	0.79	0.74	1.19	2.07	1.16	2.62	1.67	1.81	0.92
F	0.83	1.78	2.43	1.16	0.82	0.56	1.03	0.62	0.62	1.11	0.72	0.96	0.67	0.90	2.84	2.94	3.01	2.23	1.12	0.84
P	0.49	1.31	1.58	1.11	0.69	0.59	1.15	0.52	0.68	1.18	0.99	1.16	0.78	0.98	2.13	2.09	3.59	4.17	3.09	0.35
Polarity^ 1^	8.1	4.9	5.2	5.9	9.0	11.3	10.5	13.0	12.3	10.4	11.6	10.5	9.2	8.6	5.5	5.7	6.2	5.4	5.2	8.0
Hydrophobicity^ 1^	0.62	1.06	1.38	1.08	0.48	−1.5	−2.53	−0.9	−0.74	−0.4	−0.78	−0.85	−0.18	−0.05	0.29	0.64	0.26	0.81	1.19	0.12
AR_w_/AR_s _^2^	1.22	1.85	2.08	1.94	0.95	0.66	0.74	0.73	0.71	1	0.72	0.75	0.81	0.94	2.5	1.62	1.24	1.67	2.11	0.76

^1^ Data quoted from [45]. Shaded cells represent polarity score ≤ 7 or hydrophobicity score ≥ 0; 2 ARw/ARs scores are from Table 1. ARw/ARs ≥1 (shaded, row 24) represent residues that tend to distribute inside protiens; RAIR ≥ 1 (shaded, row 2–21) represent residue pairs that have high binding propensity.

**Table 3 ijms-21-00467-t003:** Comparison with existing PPIs site predictors.

Methods	AUC (%)
PSIVER [35]	62.8
PPiPP [48]	72.9
SSWRF [54]	72.9
DLPred [19]	81.1
PAIPred [24]	86.2
OURS	91.2

**Table 4 ijms-21-00467-t004:** Comparison of RFPP with PAIRPred.

Data Set	Method		RFPP					
			10%	25%	50%	75%	90%	100%
**DBD 4.0** **(116 Dimers)**	PAIRPred		1	4	11	53	194	2861
**DBD 4.0** **(116 Dimers)**	OURS	High propensity	2	8	26	69	169	580
**DBD 5.0** **(138 Dimers)**	OURS	High propensity	2	8	30	82	224	582

**Table 5 ijms-21-00467-t005:** Numbers of complexes and dimers in DBD 4.0 and 5.0.

DBD Version	No. of Complexes (Dimers)	
	Total	Used
4.0	175 (117)	174 ^1^ (116 ^2^)
5.0	230 (139)	(138 ^3^)

^1^ Used for statistics of binding propensity and distribution tendency of residues; ^2^ Used for model comparison; ^3^ Used for validation.

**Table 6 ijms-21-00467-t006:** 138 dimers used in this study.

Version	PDB ID
DBD 4.0 And DBD5.0	1ACB 1AK4 1ATN 1AVX 1AY7 1B6C 1BKD 1BUH 1BVN 1CGI 1CLV 1D6R 1DFJ 1E6E 1E96 1EAW 1EFN 1EWY 1F34 1F6M 1FC2 1FFW 1FLE 1FQ1 1FQJ 1GCQ 1GHQ 1GL1 1GLA 1GPW 1GRN 1GXD 1H1V 1H9D 1HE1 1HE8 1I2M 1IBR 1IRA 1J2J 1JIW 1JK9 1JTG 1KAC 1KTZ 1KXP 1KXQ 1LFD 1M10 1MAH 1MQ8 1NW9 1OC0 1OPH 1PPE 1PVH 1PXV 1QA9 1R0R 1R6Q 1R8S 1S1Q 1SBB 1SYX 1T6B 1TMQ 1UDI 1US7 1WQ1 1XD3 1XQS 1Y64 1YVB 1Z0K 1Z5Y 1ZHH 1ZHI 1ZM4 2A5T 2A9K 2ABZ 2AJF 2AYO 2B42 2BTF 2C0L 2CFH 2FJU 2G77 2H7V 2HLE 2HQS 2HRK 2I25 2I9B 2IDO 2J0T 2J7P 2NZ8 2O3B 2O8V 2OOB 2OT3 2OUL 2OZA 2PCC 2SIC 2SNI 2UUY 2VDB 2Z0E 3CPH 3D5S 3SGQ 4CPA 7CEI
DBD 5.0	1JTD 2A1A 2GAF 2YVJ 3A4S 3K75 3PC8 3VLB 4H03 2GTP 2X9A 3BIW 3H2V 4M76 4FZA 4IZ7 3BX7 3DAW 3S9D 3FN1 1RKE 3F1P

**Table 7 ijms-21-00467-t007:** Max accessible surface area (ASA) of 20 amino acids.

Residue	Max ASA(A2)	Residue	Max ASA(A2)
A	106	E	194
L	164	H	184
I	169	N	157
V	142	Q	198
G	84	S	130
K	205	T	142
R	248	C	135
D	163	M	188
Y	222	F	197
W	227	P	136

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
