# Peer review of "Prediction of Protein–Protein Interaction Sites Using Convolutional Neural Network and Improved Data Sets"

_ijms, 2020, doi:10.3390/ijms21020467_

Round 1

Reviewer 1 Report

The manuscript “Prediction of protein-protein interaction sites using convolutional neural network and improved data sets” reports on the use of residue binding propensity to improve the performance of protein-protein interactions domains prediction algorithms.

The results obtained improve significantly the existing approaches and the manuscript is well written considering English grammar and style

In my opinion, the clarity of results presentation can be improved if some of the information included in materials and methods (found at the end of the manuscript) are moved in the results section, or if the relevant contend is referenced properly.

Overall, the manuscript is recommended for publication after the following minor revisions:

Line 12: use present tense

Line 14: avoid the use of AUC acronym in the abstract

Line 14-16: it is recommended to rewrite this part to improve clarity

Line 22: use plural “processes”

Line 23: delete “the”

Line 50: define ROC or refer to materials section

Line 77: the sentence “achieved better results” is too generic

Line 91 (table 1): give information about the database used or reference to materials section; define all symbols

Line 109: “shown” instead of “showed”

Line 115: give reference about DBD 5.0 or refer to materials section

Line 128: replace “that” with "data sets" for clarity

Line 138: specify the comparison

Line 168: “introducing binding propensity” (add “binding” for clarity)

Line 189: replace “possible” with “likely”

Line 222: the couples should be G-P and A-N like in the picture

Author Response

Dear reviewer,

Thank you very much for your constructive comments. We have made changes accodingly as follows.

Point 1: In my opinion, the clarity of results presentation can be improved if some of the information included in materials and methods (found at the end of the manuscript) are moved in the results section, or if the relevant contend is referenced properly. 

Response 1: To make results part more readable, we have added a few sentences at the beginning of each result section (line 83, 97, 123, 139, 158, 176).

Overall, the manuscript is recommended for publication after the following minor revisions:

Point 2: Line 12: use present tense

Response 2: Changed ‘proposed’ to ‘propose’ (line 12) and ‘used’ to ‘use’ (line 12).

Point 3: Line 14: avoid the use of AUC acronym in the abstract

Response 3: Changed “AUC” to “the area under curve (AUC)”.

Point 4: Line 14-16: it is recommended to rewrite this part to improve clarity

Response 4: Rewrote line 14-16.

Point 5: Line 22: use plural “processes”

Response 5: Changed “process” to “processes” in line 10 and line 24

Point 6: Line 23: delete “the”

Response 6: Deleted “the” from “the growth and proliferation of cells” in line 24

Point 7: Line 50: define ROC or refer to materials section

Response 7: Changed “ROC” to “receiver operating characteristic (ROC)”.

Point 8: Line 77: the sentence “achieved better results” is too generic

Response 8: Deleted “and achieved better results”, and elucidated “better results” in the following sentence.

Point 9: Line 91 (table 1): give information about the database used or reference to materials section; define all symbols

Response 9: Rearranged Result section “2.1. Residue binding propensity” into section “2.1 Distribution tendency of residues in proteins” and “2.2. Residue binding propensity”, and Materials and Method section “4.3. Distribution tendency of residues in proteins” into section “4.3. Distribution tendency of residues in proteins” and section “4.4. Binding propensity of residue pairs”. Changed following section numbers accordingly.

Gave reference to section 4.3 in line 85, and define symbols in Table 1 at the table note (line 93-95).

Point 10: Line 109: “shown” instead of “showed”

Response 10: Changed “showed” to “shown” in line 119.

Point 11: Line 115: give reference about DBD 5.0 or refer to materials section

Response 11: There was a reference ([53]) about DBD 5.0. We added a reference to section 4.1 at the end of this sentence (line 127-128).

Point 12: Line 128: replace “that” with "data sets" for clarity

Response 12: Replaced “that” with “data sets” in line 141.

Point 13: Line 138: specify the comparison

Response 13: Changed “Comparison with randomly sampled data set” to “Comparison of AUC scores between data sets with high binding propensity and data sets randomly sampled.” in line 153-154.

Point 14: Line 168: “introducing binding propensity” (add “binding” for clarity)

Response 14: Added “binding” in line 199.

Point 15: Line 189: replace “possible” with “likely”

Response 15: Replaced “possible” with “likely” in line 221.

Point 16: Line 222: the couples should be G-P and A-N like in the picture

Response 16: Changed “G-W” and “A-S” into “G-P” and “A-N” in line 270.

Best regards!

Reviewer 2 Report

In their manuscript authors proposed a novel statistics-based method for the prediction of protein-protein interactions, exploiting the binding propensity of amino acids and other physicochemical properties by using convolutional neural network.

Skipping on the novelty of the approach that not appears so innovative, the first thing I'm waiting from a similar work is the availability of the code or the link to a website. In absence of these tools it is impossible to evaluate the robustness of the work, or replicate the results.

The authors not clearly described the used data-set, they refer to DBD 5.0 with 138 proteins, after to the DBD 4.0 sharing 116 proteins with DBD 5.0, and finally to a 173 protein complexes from DBD 4.0. In fact, each figure from 1 to 3 is obtained from a different data set. Therefore, it is not clear why they mixed the two databases and not use only the last released DBD 5.0. Moreover, the authors compared their methodology with just another software, this is not enough to demonstrate the robustness of their work. Also, the method appears less performing on DBD 5.0, and the manuscript lacks in the data analysis of PAIRPred using the DBD 5.0. Finally, it is not clear how authors approach to the exception in residues (such as arginine) behavior with respect to the general propensity to bind. For these reasons I suggest a major revision of manuscript.

minor comments:

Area under the ROC curve (AUC) lacks in recall, precision and accuracy values.

Please uniform in the introduction the AUC score values obtained for the different softwares for a fast and clear comparison of data.

Linee 102-103 - Please describe better the differences between arginine and histidine, which have negative hydrophobicity but high binding propensity, and alanine, glycine, and proline, which have positive hydrophobicity but low binding propensity.

line 145 modify figure number from 6 to 3.

Lines 228-230 appears to be in a wrong contex.

Correct “sturcture” in “structure” in figure 5

Check for English language.

Author Response

Dear reviewer,

Thank you very much for your constructive comments. We have made changes accodingly as follows.

In their manuscript authors proposed a novel statistics-based method for the prediction of protein-protein interactions, exploiting the binding propensity of amino acids and other physicochemical properties by using convolutional neural network. 

Point 1: Skipping on the novelty of the approach that not appears so innovative, the first thing I'm waiting from a similar work is the availability of the code or the link to a website. In absence of these tools it is impossible to evaluate the robustness of the work, or replicate the results. 

Response 1: We put the code of our paper on GitHub (https://github.com/Xiaoya-Deng/PPI-sites-prediction) and gave reference in line 314.

Point 2: The authors not clearly described the used data-set, they refer to DBD 5.0 with 138 proteins, after to the DBD 4.0 sharing 116 proteins with DBD 5.0, and finally to a 173 protein complexes from DBD 4.0. In fact, each figure from 1 to 3 is obtained from a different data set. Therefore, it is not clear why they mixed the two databases and not use only the last released DBD 5.0.

Response 2: We revised section 4.1 data sets and added a table (Table 5) to make it more clear. We also corrected a number (173 to 174) in the paper. We had miscounted 1N8O in the previous version. Sorry about that!

Point 3: Moreover, the authors compared their methodology with just another software, this is not enough to demonstrate the robustness of their work. Also, the method appears less performing on DBD 5.0, and the manuscript lacks in the data analysis of PAIRPred using the DBD 5.0.

Response 3: We added four prediction methods for comparison in section “2.5 Comparison with existing methods” (line 158-162, Table 3), and added a paragraph in Discussion to explain why we only compared the two methods on DBD 4.0 (line 188-191).

Point 4: Finally, it is not clear how authors approach to the exception in residues (such as arginine) behavior with respect to the general propensity to bind. For these reasons I suggest a major revision of manuscript.

Response 4: We added some details to the discussion of exceptional residues to explain the possible reasons in line 210, 215-216.

minor comments:

 Point 5: Area under the ROC curve (AUC) lacks in recall, precision and accuracy values.

Response 5: We have added accuracy and recall under different thresholds in Table S3 in Supplementary Material and given reference in line 134.

Point 6: Please uniform in the introduction the AUC score values obtained for the different softwares for a fast and clear comparison of data.

Response 6: We have changed AUC scores in percent format to decimals in introduction section.

Point 7: Line 102-103 - Please describe better the differences between arginine and histidine, which have negative hydrophobicity but high binding propensity, and alanine, glycine, and proline, which have positive hydrophobicity but low binding propensity.

Response 7: We changed the sentence to explain in more detail the differences between the two kind of exceptions in line 112-114.

Point 8: line 145 modify figure number from 6 to 3.

Response 8: Changed figure number 6 to 3.

Point 9: Lines 228-230 appears to be in a wrong contex.

Response 9: To make this part more readable, we have rearranged Result section “2.1. Residue binding propensity” into section “2.1 Distribution tendency of residues in proteins” and “2.2. Residue binding propensity”, and Materials and Method section “4.3. Distribution tendency of residues in proteins” into section “4.3. Distribution tendency of residues in proteins” and section “4.4. Binding propensity of residue pairs”. Changed following section numbers accordingly.

Point 10: Check for English language.

Response 10: We have checked the whole paper and made some changes in line 9, 10, 12, 13, 24, 65, 119, 221.

Best regards!

Round 2

Reviewer 2 Report

Unfortunately, they did not added the modified text in the answers, so I cannot give a complete judgment on their work, however they made the code of their software available on GitHub and added the link in the text. This is the most important aspect of their work, which demonstrate their experimental activities and give the opportunity to the scientific community to use this tool and judge how much could be useful. Moreover, I suggest to remember authors to cite their paper (if published) also in their GitHub site using a sentence like: "If you publish pictures or models using our software please cite the following paper: ....", to publicize the journal.

Anyway, they answered point by point to all questions, so in general I think they sure improved their manuscript and I agree with the acceptance for publication. I will read the article when published.